Susceptibility to experimental infection of the invertebrate locusts (Schistocerca gregaria) with the apicomplexan parasite Neospora caninum

Alkurashi Mamdowh M. 1 2
May Sean T. 3
Kong Kenny 4
Bacardit Jaume 5
Haig David 1
Elsheikha Hany M. 1 hany.elsheikha@nottingham.ac.uk
1 School of Veterinary Medicine and Science, University of Nottingham , Sutton Bonington Campus, Leicestershire , UK
2 Animal Production Department, College of Food and Agricultural Sciences, King Saud University , Riyadh , Saudi Arabia
3 Nottingham Arabidopsis Stock Centre, Division of Plant and Crop Sciences, School of Biosciences, University of Nottingham , Leicestershire , UK
4 School of Physics and Astronomy, University of Nottingham , UK
5 The Interdisciplinary Computing and Complex BioSystems (ICOS) Research Group, School of Computing Science, Newcastle University , Newcastle-upon-Tyne , UK
Kumar Abhishek
Electronic publication date: 2014 Dec 2
Publication date: 2014
Volume: 2
Electronic Location ID: e674
Received 2014 Jul 11; Accepted 2014 Nov 1
Copyright: © 2014 Alkurashi et al.
Copyright year: 2014
Copyright holder: Alkurashi et al.
License: This is an open access article distributed under the terms of the Creative Commons Attribution License, which permits unrestricted use, distribution, reproduction and adaptation in any medium and for any purpose provided that it is properly attributed. For attribution, the original author(s), title, publication source (PeerJ) and either DOI or URL of the article must be cited.
License URL: https://creativecommons.org/licenses/by/4.0/

Keywords: Infection, Host-pathogen interaction, Locusts, Behaviour, Invertebrate model

Funding: Studentship from the Ministry of Higher Education in Saudi Arabia Mamdowh Alkurashi was funded by Ministry of Higher Education of Saudi Arabia. The funder had no role in study design, data collection and analysis, decision to publish, or preparation of the manuscript.

==============================
Neuropathogenesis is a feature of Neospora caninum infection. In order to explore this in the absence of acquired host immunity to the parasite, we have tested infection in locusts (Schistocerca gregaria). We show for the first time that locusts are permissive to intra-hemocoel infection with N. caninum tachyzoites. This was characterized by alteration in body weight, fecal output, hemoparasitemia, and sickness-related behavior. Infected locusts exhibited progressive signs of sickness leading to mortality. Also, N. caninum showed neuropathogenic affinity, induced histological changes in the brain and was able to replicate in the brain of infected locusts. Fatty acid (FA) profiling analysis of the brains by gas chromatography and multi-variate prediction models discriminated with high accuracy (98%) between the FA profiles of the infected and control locusts. DNA microarray gene expression profiling distinguished infected from control S. gregaria brain tissues on the basis of distinct differentially-expressed genes. These data indicate that locusts are permissible to infection with N. caninum and that the parasite retains its tropism for neural tissues in the invertebrate host. Locusts may facilitate preclinical testing of interventional strategies to inhibit the growth of N. caninum tachyzoites. Further studies on how N. caninum brings about changes in locust brain tissue are now warranted.

Introduction

Infection by the apicomplexan protozoan Neospora caninum is an important cause of infertility and abortion in cattle and neuromuscular disorders in dogs (Barber & Trees, 1996; Vonlaufen et al., 2002; Dubey, Schares & Ortega-Mora, 2007; Innes, 2007). With no effective vaccine and limited anti-parasitic treatments available, N. caninum poses a pressing economical and animal health threat. Limited understanding of molecular mechanisms of cerebral neosporosis in hosts contributes to the lack of effective treatments. Hence, increased knowledge of the molecular events that lead to brain damage is essential.

Neospora caninum is an obligatory intracellular organism that has the capacity to invade a wide range of cells both in vitro and in vivo (Vonlaufen et al., 2002). Various infection hosts have been used to understand N. caninum neuropathogenesis including in vitro studies using cell lines or organotypic cultures, which provided key information about host cell invasion by N. caninum and the cellular events that occur during intracellular parasite proliferation (Vonlaufen et al., 2002; Dubey, Schares & Ortega-Mora, 2007). Studies of N. caninum infection and the resulting neuropathologies have also been observed for in vivo vertebrate models, including: cats, mice, rats, gerbils and monkeys (Dubey, 1999; Collantes-Fernández et al., 2004; Pinheiro et al., 2006; Bartley et al., 2007; Reichel & Ellis, 2009). These models suggested several molecules and mechanisms key to the establishment and progression of infection (Collantes-Fernández et al., 2002; Vonlaufen et al., 2002), and allowed potential vaccinations for neosporosis to be assessed (Bartley et al., 2007).

Vertebrate animals are expensive to maintain and large numbers are usually needed to yield statistically valid data. Thus, understanding the molecular determinants of N. caninum brain infection in a cost effective and productive organ culture or whole animal system would be advantageous. Virtually nothing is known about invertebrate host responses to infection with N. caninum. Invertebrate hosts have proven valuable in understanding aspects of host-parasite interactions (Siddiqui et al., 2011). For example, the protist, Acanthamoeba shows commonality in targeting the blood–brain barrier of locusts and humans, suggesting that the invertebrate locust can mimic human Acanthamoeba encephalitis and reveal a spectrum of host-pathogen interactions (Mortazavi et al., 2009; Mortazavi et al., 2010). Examples such as these support the use of a locust system to study protozoan infection and to investigate virulence determinants in vivo.

Insects rely on an innate immune system for their protection against infection, so the use of an insect model is relevant in the study of N. caninum infection, the control of which has dependency on the innate immune system (McAllister et al., 2000; Bartley et al., 2013). The evolutionary conservation of several aspects of the innate immune response between invertebrates and mammals (Hoffmann et al., 1999; Lemaitre & Hoffmann, 2007) and the fact that N. caninum utilizes similar virulence factors in phylogenetically distant hosts studied so far makes the use of a simple invertebrate host such as locust an attractive one for studying parasite pathogenesis. Indeed, locusts have already been used to answer questions regarding host-pathogen interactions in other microbial and protozoan systems, such as Acanthamoeba (Mortazavi et al., 2009; Mortazavi et al., 2010) as mentioned above and also in studies with neuropathogenic E. coli K1 (Khan & Goldsworthy, 2007; Mokri-Moayyed, Goldsworthy & Khan, 2008).

In this study, we examined the susceptibility of a non-mammalian host for infection with N. caninum. Adult locusts, Schistocerca gregaria are found to be susceptible to infection with N. caninum tachyzoites given via the intra-hemocoel route of inoculation and can be used as surrogate hosts for N. caninum. The parasite infection exhibited strict neurotropism, and resulted in brain pathology and molecular changes associated with sickness and ultimately death of the locusts.

Materials and Methods

Research ethics statement

This study was reviewed by the University of Nottingham (UK) School of Veterinary Medicine and Science (SVMS) Ethical Review Committee. The Committee reviews all research studies involving School personnel and is chaired by Professor David Haig. The committee passed this study as good to proceed, not requiring any further ethical review as it involved invertebrates. FELASA guidelines as outlined in ‘principles and practice in ethical review of animal experiments across Europe (2005)’ and UK guidelines on the use of invertebrates in research were followed.

Parasite strain and growth conditions

Neospora caninum (Liverpool strain) was cultured in human brain microvascular endothelial cells (HBMECs) originally obtained from ScienCell Research Laboratories (Elsheikha et al., 2013). HMBECs were used at passage 18 and were grown in T75 flasks (Corning) in tissue-culture medium composed of RPMI-1640 medium containing L-glutamine and sodium bicarbonate, and supplemented with 20% (v/v) heat inactivated FBS, 1 mM Sodium Pyruvate, 1% MEM non-essential amino acids, 1% MEM vitamins and 2% penicillin/streptomycin. Cells were maintained in an incubator in a humidified atmosphere at 37 °C and 5% CO2. Once confluent (∼3 days), cells were trypsinized using trypsin-EDTA (Invitrogen, GIBCO, UK). N. caninum tachyzoites were harvested from cultured cells, purified as described (18), and counted using a haemocytometer and diluted to the desired concentration, 103, 104, 105 and 106 tachyzoites suspended in 20 µl culture medium, for inoculation (Elsheikha et al., 2006).

Locusts

Desert locusts, Schistocerca gregaria of gregarious morphotype, were obtained from a local breeder (www.livefoodsdirect.co.uk). Locusts were kept in a large glass cage for not less than one week after arrival to acclimatize and reduce any stress associated with transportation. Adult locusts were used throughout the study to ensure a fully developed innate immune system. Each group of 10 locusts (5 males and 5 females) were housed in separate plastic “critter cages” with ventilation slats in the lid and solid plastic bottoms to enable feces collection. They were fed daily with fresh grass and wheat seedlings supplemented with bran. Water was provided ad libitum in petri dishes to increase hemolymph volume. The “critter cages” were kept on the bench in a controlled environment at 25 °C and 65% humidity with a 12-h light cycle. During the parasite exposure period, housing and manipulation of locusts was performed within a biological safety cabinet (class II) inside an ABSL2 + containment area.

Infection conditions

Locusts were randomized into five groups (G1 to G5), with 10 locusts per group and were housed in separate “critter cages” to prevent cross-cage contamination after parasite challenge. To establish a suitable dose for N. caninum tachyzoites, 103, 104, 105, or 106 tachyzoites were inoculated in a total volume of 20 µl culture medium into the hemocoel of locusts in groups G1 to G4, respectively. Locusts in G5 were mock-inoculated with medium only and used as controls. An additional control group was included (i.e., locusts which were not injected but incubated under the same conditions as infected locusts). Injection of the test materials was achieved using a 26g Terumo® Neolus hypodermic needle fitted into plastic pipette tips. Inocula were injected into the hemocoel by inserting the needle horizontally into the inter-segmental membrane at the last two abdominal segments using a pipetter, followed by gentle stretching of the locust’s abdomen to ensure an even distribution of parasites within the hemocoel (Fig. S1). Individual locusts were identified by numbering on the underside of the thoracic cuticle using a permanent marker pen. Locusts were observed daily for the duration of the experiment.

Survival

Locusts were assessed for mortality twice daily and any dead locusts were removed to prevent cannibalism. Post mortem analysis was performed within the safety cabinet. Control locusts were sacrificed by making an incision behind the head at the end of the experiment. Survival data was analyzed using the Gehan-Breslow-Wilcoxon test, with death as the primary variable using GraphPad Prism version 5.00 for Windows (GraphPad Software, San Diego, CA). Significance between survival curves was assessed using the Log-rank (Mantel-Cox) test.

Morbidity after parasite challenge

To assess the effect of N. caninum infection on the behavior of the locusts, a range of parameters were evaluated. These include the following: (i) The number of times the insect was inactive. (2) The number of walking bouts (a walk was defined as an unbroken period of locomotion of more than half a body length). (3) The number of times leg movements occurred. (4) The number of times antennal movements occurred. (5) The number of times grooming occurred. (6) The number of times wing-fanning occurred. (7) The number of times jumping occurred. This experiment was divided into two 15 min periods. During the first 15 min period, individual behavioral variables, chosen after preliminary observations, were registered every 10 s for locusts in each cage.

Detection of hemoparasitemia

To determine whether locusts developed parasitemia, hemolymph was collected from one locust per group 3 h PI and daily thereafter for three consecutive days. Hemolymph was collected from locusts by inserting a 1–10 µl pipette tip fitted into a pipette-aid into the arthrodial membrane at the base of the legs (Fig. S2). At least 20 µl of hemolymph was aspirated from each locust (depending on the hydration of the locust) and a smear was made on a clean microscope glass slide and checked for the presence of N. caninum tachyzoites under X40 magnification.

Body weight

To assess the effect of N. caninum infection on food consumption locusts were weighed prior to infection and daily for the duration of the experiment. The body weight changes were calculated as a percentage of initial body weight. Each locust was transferred from the cage within a closed 50-ml falcon tube, weighed on a weighing balance and returned back to the cage within the same falcon tube. Results are presented as means ± SEM from three independent experiments.

Fecal output

Another indirect measure was used to assess the effect of exposure to N. caninum on food consumption where the output of fecal pellets was determined on a daily basis. Fecal pellets produced per cage over a 24 h period were separated from non-fecal materials and weighed. This figure was divided by the number of surviving locusts to give an average of fecal output per locust.

Detection of N. caninum in locusts’ brain

In a second experiment one group of five locusts were inoculated with 104 tachyzoites and another group was sham inoculated with medium only. One locust from each of the infected and control groups was sacrificed daily and samples from the brains of each locust were collected and preserved in 5% paraformaldehyde for histology or 70% ethanol for genetic characterization. For histological analysis, samples were fixed for at least 24 hr before they were sectioned and stained with hematoxylin and eosin. Also, PCR-DNA sequencing was used to document the presence of growing parasites in the brain. Genomic DNA was extracted from the locust’s brain using a DNeasy Blood and Tissue Kit (Qiagen Inc.) following the Animal Tissues Spin Protocol supplied by the manufacturer. The DNA was eluted in 50 µl of kit elution buffer and stored at −20 °C. Concentration and purity of DNA in sample extracts were checked by using a Thermo Scientific NanoDrop™ 1,000 Spectrophotometer, prior to PCR amplification. The two most commonly species-specific primers used to diagnose N. caninum, primers Np6 plus (5′CTCGCCAGTCAACCTACGTCTTCT3′) and Np21 plus (5′CCCAGTGCGTCCAATCCTGTAAC3′), were used to amplify approximately 337-350-bp fragment of the Nc5 gene, which is commonly used to identify N. caninum infection (Muller et al., 1996). Nc5 has been shown in previous studies to be N. caninum species-specific and a variable region (Kaufmann et al., 1996; Yamage, Flechtner & Gottstein, 1996). Hence, this gene might be a useful genetic marker for discrimination between different N. caninum isolates. Approximately 50 ng of genomic DNA was amplified using 40-µl reaction mixture containing 2 µl of extracted DNA, 20 µl of Biomix (Bioline, UK), 17 µl of nuclease-free water (Fisher Scientific), and 0.5 µl (∼10 pmol) of each forward and reverse oligonucleotide primer (Eurofins). BioMix™ is a complete, ready-to-use, 2 × reaction mix containing Taq DNA polymerase. It is used to perform PCR assays of numerous genomic and cDNA templates by only adding water, template and primers. All amplifications were carried out in triplicates using the Bioer Xp Cycler. The PCR cycling conditions consisted of an initial denaturation for 5 min at 94 °C and then 40 cycles of 94 °C for 1 min, 63 °C for 1 min, and extension at 74 °C for 3.5 min, followed by a final extension at 72 °C for 10 min. A negative control (nuclease-free water instead of DNA) and a positive control (DNA from cultured-derived N. caninum) were included in each run. The quality and specificity of all the amplification products were assessed by 2% agarose gel electrophoresis followed by staining with ethidium bromide. Individual PCR product bands were visualized under ultraviolet light. Amplicons were purified using a QIAquick PCR Purification Kit (Qiagne) according to the manufacturer’s instructions. Cleaned-up products were sequenced bidirectionally using the same PCR oligonucleotide primers used in initial amplification using a commercial service (Source Biosciences Inc., Nottingham, England). To gain insight into the anatomic sites of N. caninum distribution and replication, we detected N. caninum DNA in several different body tissues of locusts using the same PCR assay described above.

Recovery of infecting parasite isolate

For this experiment 12 freshly dead locusts were randomly selected from groups which have been infected with 103, 104, 105, or 106 tachyzoites (three locusts per group). At 5, 10 or 17 day PI brain was obtained from one locust from each group (i.e., a total of four brains, one from each group), and were dissected under aseptic conditions for parasite isolation. Brains were washed twice in RPMI medium, divided into small pieces by scissors and homogenized by pipetting up and down gently 10 times through a 1-ml pipette tip, followed by passing the homogenate through an 18-gauge needle and syringe. The resulting homogenate from four brains at each of the time points mentioned above were pooled and resuspended in 6 ml of RPMI medium. The 6 ml homogenate from the pooled four brains (1ml/well) was overlaid on healthy HBMECs seeded in a 6-well culture plate (Nunc Inc., Denmark) and topped up by a second ml of fresh RPMI medium, and plate was incubated at 37 °C. After 24 h the culture medium was decanted, and 2 ml of fresh RPMI medium was added to each well. One 6-well culture plate was seeded at 5, 10 or 17 day PI. All plates were incubated in a humidified atmosphere at 37 °C, and were examined daily for signs of parasite growth within cultured cells by inverted light microscopy.

Characterization of locust’s brain-derived isolate

We tested if the passage of N. caninum in a non-natural host has favoured parasite phenotypic or genotypic changes. The species identity of the locust-derived N. caninum isolate was confirmed and characterised in comparison to the original isolate. Tachyzoites from each isolate were subjected to a range of phenotypic and genetic characterisation methods. Firstly, immunofluorescence staining of tissue culture infected with each isolate was performed as described previously (Alkurashi et al., 2011). Secondly, for transmission electron microscopy (TEM), HBMEC culture containing tachyzoites originated from each isolate were fixed in 2% glutaraldehyde in 0.2 M sodium cacodylate buffer (pH 7.4) for 3 h and were post-fixed for 1 h in 1% osmium tetroxide in the same buffer; they were then dehydrated in acetone and embedded in epoxy resin. Finally, ultrathin (80 nm) sections were cut with a Leica EM UC6 microtome, and contrasted with uranyl acetate and lead citrate. Sections were examined with a Phillips Morgagni 268 (FEI company, Hillsboro, OR) transmission electron microscope operating at 80 kV. Digital images were recorded with a MegaViewIII digital camera operated with iTEM software (Olympus Soft Imaging Systems, Germany). The Adobe Photoshop CS4 digital photography editing program was used for additional processing. Thirdly, purified parasite preparation from each isolate was used for PCR-sequencing analysis using species-specific primers Np21plus–Np6plus which anneal into the Nc5 region of N. caninum as described above. Fourthly, chemical profile of tachyzoites from each isolate was obtained by using Confocal Raman spectroscopy, scanned over at 0.5 µm step sizes. At least 13 tachyzoites from each isolate were studied. A k-means clustering method was used to separate the spectra from the tachyzoites and substrate, and then, the spectra from tachyzoites of both isolates were used to do principle component analysis (PCA) to identify chemical differences. The measurements were performed using 785 nm laser, and at 3 s integration time for every spectrum. Fifthly, approximately 105 tachyzoites derived from one locust-derived isolate were used for subsequent inoculation into five new locusts to assess the effect of passage into a non-natural host on the parasite’s ability to retain its neurotropism. Locusts were monitored daily for 3 weeks for mortality and sickness behavior. All experiments were performed three independent times.

Gas Chromatography (GC) profiling of brain lipids

Twenty five locusts were randomly allocated to five groups each with five locusts. Each locust was inoculated with 104 N. caninum tachyzoites; this dose was chosen because it produces clinical illness in the infected locusts. Negative-control locusts were injected with 20 µl of medium. One day after infection and daily thereafter, five locusts from each group (infected or control) were sacrificed by cervical dislocation and their brains were dissected within the safety cabinet, providing five replicates per group. Brain samples were immediately frozen in liquid nitrogen and homogenised using a gentleMACSTM closed homogeniser (Milteny Biotec Ltd., Surrey, UK). Homogenised tissues were subjected to lipid extraction as described previously (Folch, Lees & Sloane Stanley, 1957). Samples were transesterified by the method of Christie (1982) and modified by Chouinard et al. (1999). The fatty acid methyl esters were injected (split ratio 50:1) into gas chromatograph (GC 6890; Agilent technologies Ltd, Stockport, UK). Separation of fatty acid methyl esters was performed with a Varian CP-Sil 88 (Crawford ScientificTM Ltd., Strathaven, UK) capillary column with hydrogen as carrier gas. The fatty acid methyl esters were identified by comparing the retention times with a fatty acid methyl esters standard mixture (Sigma-Aldrich Co LLC, Gillingham, UK) and the area percentage in moles were used for the statistical analysis. A total of 37 fatty acids were analyzed in this study, and included: saturated fatty acids: C4:0, C6:0, C8:0, C10:0, C11:0, C12:0, C14:0, C15:0, C16:0, C17:0, C18:0, C20:0, C21:0, C22:0, C23:0, C24:0. Monosaturated fatty acids: C14:1, C15:1, C16:1, C17:1, C20:1, C24:1. Polysaturated fatty acid- Omega (n)-3: C18:3n3, C20:3n3, C20:5n3, C22:6n3. Omega (n)-6: C18:2n6t, C18:2n6, C20:3n6, C20:4n6. Omega (n)-9: C18:1n9t, C18:1n9C, C20:1n9.

Data obtained from the brain phospholipid composition was filtered of poorly measured fatty acids (defined as fatty acids with percentiles below 0.01%). Afterwards, the data were evaluated using SPSS 16.0 (SSPS® Inc., Chicago, USA) using a general linear model. The significance of each fatty acid was assessed using p-values corrected for multiple comparisons using the Bonferroni method. All fatty acids that obtained a p-value <0.05 were kept for further analysis.

Multi-variate prediction models were constructed to determine if it was possible to discriminate between the parasite-infected samples and the controls. Specifically we have used our own machine learning method called BioHEL (Bacardit, Burke & Krasnogor, 2009), that has been successfully applied in the past for the analysis of lipidomics data (Fainberg et al., 2012) and transcriptomics (Bassel et al., 2011; Glaab et al., 2012). The prediction capacity of BioHEL on the locust samples was estimated using the robust leave-one-out cross-validation procedure, in which a prediction model is built using all but one of the samples, and the remaining sample is used to test the validity of the model. This procedure is repeated 50 times, each time using a different sample to test the models. The prediction capacity of the system is estimated as the percentage of correctly predicted samples.

Finally, in order to identify which are the combinations of fatty acids presenting the highest prediction capacity, new prediction models were generated where each model used only two, three or four of the fatty acids, testing all possible combinations. Again, leave-one-out cross-validation was used to estimate the prediction capacity of these reduced panels of fatty acids.

Detection of differential gene expression using the Drosophila gene chip

In a new experiment three locusts were each inoculated with 104 N. caninum tachyzoites. Three locusts were sham inoculated with RPMI medium and used as controls. One day after infection, locust brains were dissected, frozen in liquid nitrogen, transferred to a pre-chilled (with liquid nitrogen) mortar and ground with a pestle to a very fine powder with liquid nitrogen. The powdered samples were transferred to 2 ml Eppendorf microtubes and used for extraction of total RNA. Total RNA was extracted from 50 mg sample powder using the QIAGEN RNeasy Mini Kit (QIAGEN, Germantown, MD). To verify the quality of the RNA, the yield and purity were determined spectrophotometrically (NanoDrop, Wilmington, DE) and with an Agilent 2100 Bioanalyzer using RNA 6000 Nano kits (Agilent Technologies, Palo Alto, CA). A whole transcript drosophila array (GeneChip Drosophila Genome 2.0 Array; Affymetrix, Santa Clara, USA) was assessed for the study of gene expression in locust. The hybridization of heterologous (non-specific) nucleic acids onto arrays designed for model-organisms have been shown to be a viable genomic resource for estimating gene expression in many non-model organisms (Lai, May & Mayes, 2014). This approach enables the investigation of gene expression in locusts in the absence of a fully sequenced genome.

Analysis of the cross-species hybridization data was performed as follows: Biotin labeled locust genomic DNA was hybridized to a drosophila_2 chip (Affymetrix, Santa Clara, USA) and used as described previously (Lai, May & Mayes, 2014) to convert a standard drosophila_2 Chip Definition File (CDF file; Affymetrix, Santa Clara, USA) into a series of bespoke locust.cdf files. The newly derived CDF file having the maximum number of filtered probes, whilst retaining all probe-sets, was selected and subsequently used in all downstream analysis. Statistical analysis of mRNA expression profiling was done by analysis of variance (ANOVA) using Partek Genomics Suite 6.5 (Partek Incorporated, MO). Results were significant if fold change was more than 1.5 or less than −1.5, with P < 0.05. Cluster analyses and principal component analysis were conducted with Partek default settings.

Results

Clinical outcomes of infection

Mortality rate

Locusts died gradually as the infection progressed over the course of the experiment which lasted up to 27 days PI. Locusts infected with larger number of N. caninum tachyzoites showed less survival time, where all locusts in G1, G2, G3 and G4 died by day 25, 23, 21 and 20 PI, respectively (Fig. 1). Compared with control locusts, locusts receiving N. caninum infection appeared to have statistically significant higher mortality [control vs. (G1, p = 0.0008), (G2; p = 0.0007), (G3; p = 0.0004), or (G4; p = 0.0047)]. The mortality rate didn’t follow a dose-dependent manner, despite the increase in inoculation dose. Two locusts from the negative (uninfected) control group and one locust from the environmental control group had died during the course of the experiment, but their death was not due to infection with N. caninum based on a negative PCR result for brains of those three locusts.

Figure 1 Survival of locusts given various doses of Neospora caninum tachyzoites by the intra-hemocoel route.

Groups (G1 to G4) of locusts (n = 10) were administered doses of N. caninum of 103 (G1), 104 (G2), 105 (G3), and 106 (G4) per locust. Control locusts were sham-inoculated with RPMI cultured medium. An environmental control group (e-group) of non-infected locusts incubated under the same conditions as other groups was also included. Survival was monitored daily after infection. Results represent average survival curve based on three independent experiments. Control vs. G1 (p = 0.0008); control vs. G2 (p = 0.0007); control vs. G3 (p = 0.0004); control vs. G4 (p = 0.0047). Locusts inoculated with 10 or 100 tachyzoites did not exhibit any signs of sickness or mortality (data not shown).

Effects of infection on sickness behavior

Assessment of various behavioral parameters revealed significant inter-locust variability that did not correlate with differences in the dose of infection. For example, variability in behavioral parameters in individual locusts reared in the same cage varied by as much as 5-fold, which precluded any statistical analysis to be performed. The source of this variability among locusts is not understood despite the standardization of age, food, handling, and rearing conditions. Despite this variability we observed abnormal behavior in locusts two days after infection that increased over time. Most of the locusts exhibited a sluggish mobility one day before death and some showed dark blue discoloration on the day of death.

Hemoparasitemia

Tachyzoites of N. caninum were observed in the hemolymph collected from all infected locusts up to 48 h PI. Subsequently, no evidence of the parasites was obtained in hemolymph from any locust. Collection of 20 µl of hemolymph from each locust did not seem to have any effect on the activity or weight of the infected or control locusts.

Effects of infection on body weight

Infection with N. caninum did not cause a remarkable reduction in the body weight (Fig. 2). However, the distribution of data over the experiment shows that body weights decrease gradually in infected groups. At the lowest parasite challenge (G1 infected with 103 tachyzoites), the locusts did not gain or lose weight, and there was no significant difference between locusts from the sham-infected or environmental groups. However, there was a statistically significant difference in weight change between control locusts and N. caninum-infected locusts in the other groups (G2, p = 0.0024; G3, p < 0.0001; G4, p = 0.0012). The calculation of the average body weight of each group could not be performed beyond day 15 PI because only a few locusts were left in infected groups, which precluded direct quantitative comparison between different groups.

Figure 2 Relative body weight (BW) of Neospora caninum-infected locusts compared to controls at different time points after infection.

Shown are means ± SEM of percent body weight change compared with initial body weight for surviving locusts at each time point. There was no significant change in the BW of control locusts and locusts in group 1 (infected with 103 tachyzoites) along the course of the experiment, but in groups 2 and 4, infected with 104 and 106, respectively, there was significant loss in weight beginning by 2 day after infection (p = 0.0024 and 0.0012, respectively). In group 3 infected with 105 the weight loss began by 3 day after infection (p < 0.0001). Not all locusts completed the course of the experiment due to associated mortality. Data was compared using paired t-test (p-value < 0.05).

Fecal output

A slight increase in fecal output was detected in infected locusts 1 day PI, followed by progressive decrease in the subsequent days especially in locusts given the higher doses. But, a dose–response reduction in the fecal output was not detected (Fig. 3). No difference was detected in any of the above tested clinical parameters of infection between either control or infected male and female locusts (data not shown). However, there was some variability in the response of locusts, not gender-specific, from experiment to experiment.

Figure 3 The effect of Neospora caninum infection on locust fecal output.

Besides the environmental control (E-control) group, an additional group of locusts were inoculated with media only and considered the non-infected control. Groups 1, 2, 3, and 4 were infected as described in materials and methods. Fecal output per group was weighted daily for up to 7 days PI. Total fecal output was divided by the number of living locusts for every day. There was non-significant increase in fecal output one day after infection, followed by significant decrease until day 7 after infection, with p-value 0.03, 0.04, 0.05, and 0.03 for group 1, 2, 3, and 4, respectively. Data was compared using paired t-test (with p-value <0.05 as significant). Results are presented as means from three independent experiments.

Evidence for N. caninum parasites in the brain

Histological changes in the brain

Histologic examination of brain tissue from mock- and N. caninum-infected locusts demonstrated substantial changes in the brain of infected locusts. Even though the parasite was not observed in histological section there was accumulation of many inflammatory cells, located primarily in the white matter tracts of the brain of infected locusts, whereas no changes or immune response were detected in control locusts (Fig. 4). There was no significant difference of inflammatory loads observed between locusts’ brains from different infection groups. Given the significant mortality with 103 tachyzoites in group 1 it is interesting that surviving animals have normal body weight. Apart from the limited pathological changes that were detected in surviving locusts, there is no evidence to indicate that surviving locusts from this group were able to control the infection compared to those who died. However, this might be attributed to the variability in individual response of locusts to infection. The parasite has not been detected in any other organ of any locust examined.

Figure 4 Representative micrographs of Neospora caninum-infected locust brains.

Locusts were injected with 106 N. caninum and their brains were dissected out at 5 days post-infection. Subsequently, the brains were sectioned and stained with haematoxylin and eosin. N. caninum triggered inflammatory response (arrows) in the brain tissue of infected locusts (A). No parasite was detected in the brain (A) or in the fat body surrounding the brain of (B). Magnification ×400.

DNA extraction and PCR amplification

In all brain tissue of infected locusts, the PCR products were successfully amplified to the expected 337-bp parasite-specific fragment, which was found to correspond to the targeted N. caninum sequence within the Nc5 gene as confirmed by sequencing analysis. Positive PCR products from locusts’ brains were obtained from day one to day five PI (Fig. 5) and were found to slightly increase as infection progressed over time. Even though end-point PCR can detect and quantify specific DNA sequences it is commonly used as a semi-quantitative method, and thus results should be confirmed by reverse transcription-quantitative PCR (RT-qPCR). No genetic evidence for the presence of tachyzoites was detected in the brain of control locusts for up to 14 day PI. PCR was also used to determine parasite distribution to other body sites (fat body and muscle). The parasite DNA was not detected in any of the examined non-brain tissues. Likewise, there was no evidence for the presence of the parasite in feces.

Figure 5 PCR amplification of the Neospora caninum-specific Nc5 region (Np21/Np6).

Amplification of DNA extracts from brains of locusts experimentally infected with N. caninum showed the presence of genetic evidence of N. caninum in the brain of locusts from day 1 (d1) to day 5 (d5) PI. M: 100-bp molecular size marker; Lane 1: positive control represent DNA extracted from ∼3 × 106 tachyzoites; lanes d1 to d5: N. caninum in brain d1 to d5.

Recovery of viable parasites from locusts’ brains

Using a cell culture bioassay, viable N. caninum tachyzoites were successfully isolated from pooled brain homogenates of locusts that were infected with 103, 104, 105, or 106 tachyzoites at day 10 and 17 PI. Out of the 18 inoculated wells only two wells at 10 and 17 day PI yielded new tachyzoites. Recovered isolates were maintained in vitro via serial passages in cultured endothelial cells for four months before being stored in liquid nitrogen. No significant difference was observed in the rate of proliferation and growth kinetic of both locust-derived isolates and the culture-derived isolate of N. caninum (data not shown).

Characterization of N. caninum isolate derived from locusts’ brains

Immunofluorescent (Fig. S3) and TEM (Fig. S4) analyses showed that tachyzoites derived from locusts’ brains are morphologically-identical to tachyzoites of the culture-derived original isolate. Locust-derived tachyzoites retained the same phenotypic and genetic characteristics of tachyzoites of the original N. caninum isolate. DNA sequences from N. caninum tachyzoites of each isolate was identical, and a consensus sequence obtained from both isolates showed 100% sequence homology to N. caninum sequence accession number AY497045 in GenBank database.

It was important to find out if the passage of the parasite through the locust had induced any alteration in the phenotype of the parasite. Chemical profiling using Confocal Raman Spectroscopy technique (Fig. 6) and principle component analysis (Fig. 7) revealed minor, non-significant differences between the two N. caninum isolates. Locust-brain-derived N. caninum isolate was used to infect locusts to test their neuropathogenic capacity. In vivo passage in the locust’s brain did not result in alteration in neuropathogenicity. One of the locust-adapted isolates induced brain infection in 100% of five newly infected locusts, which had detectable parasite burdens in the brain at different time points PI as evidenced by PCR. But no further attempt was made to re-isolate the parasites from these locusts and use for infection again.

Figure 6 Raman spectroscopic imaging of tachyzoites.

Bright field (A) and corresponding Raman (B) image of N. caninum tachyzoites derived from locusts’ brains. Bright field (C) and corresponding Raman (D) image of N. caninum tachyzoites derived from culture. Bar applies to all figures, 5 µm.

Lipid profiling

In an effort to understand the mechanisms underlying the complex relationships between the host locust and parasite infection, we compared the fatty acid profile of brain from infected locusts to that for their control counterparts. Among the 37 fatty acids analyzed in this study, results were consistently obtained from all tested samples for only seven FAs and included: saturated fatty acids: C14:0, C16:0, C18:0; monounsaturated fatty acids: C16:1; polysaturated fatty acid-Omega (n)-3: C18:3n3. Omega (n)-6: C18:2n6, C18:1n9C. Figure 8 shows a heatmap that should be visualizing the relationships of these fatty acids to the 50 samples (25 infected, 25 controls). The heatmap suggests that no individual fatty acid is strongly associated to any of the two groups. Nonetheless, our analysis using the BioHEL machine learning algorithm reveals that we can create multi-variate prediction models that can assign samples to treatments with very high accuracy. The results of the data analysis are summarised in Table S1. When the prediction models use the seven FAs we can correctly predict 98% (all but one) of the samples. To check if an even more reduced panel of fatty acids presents high discriminative power we tested all combinations of two, three and four fatty acids. The best groups of four [Palmitic acid methyl ester (C16:0), Palmitoleic acid methyl ester (C16:1), Linoleic acid methyl ester (C18:2n6c), and Linolenic acid methyl ester (C18:3n3)] and three [C16:1, C18:2n6c and C18:3n3] FAs managed to still give an accuracy of 98%. The best group of two fatty acids (C16:1 C18:2n6c) reduced its accuracy to 94% (mis-classifying three samples).

Gene expression microarray data

To obtain a broader understanding of the effects of the N. caninum on S. gregaria, we examined the parasite impact on host gene expression using Affymetrix DNA microarray and Cross Species Hybridisation (CSH) analysis. Since N. caninum infection induced neurological injuries in locusts, it was important to test whether N. caninum infection had altered gene expression within locust brains. The analysis was designed to gain further understanding of the mechanisms for N. caninum neuropathy, and in particular, of the genes responsible for the capacity of N. caninum to establish brain infection. We used an established (Lai, May & Mayes, 2014) cross species hybridisation (CSH) approach, which applied genomic DNA pre-filtration to identify conserved probes operationally useful between phylogenetically disparate species; in this case allowing us to use Drosophila Gene Chips to assay locust RNA. Samples of RNA from brains of mock- and N. caninum-infected locusts were labeled and hybridised to Affymetrix Gene Chip Drosophila Genome 2.0 Arrays. Data were analyzed using Partek Genomics Suite Version 6.5. Of the 28,593 transcripts represented on the GeneChip, approximately 18,500 were expressed in the N. caninum-infected locust’s brain. PCA quality control analysis shows that the uninfected and infected samples were biologically noisy but generally well separated (Fig. S5). The subsequent analysis of differential expression was constrained by ≥1.5-fold difference in expression where Fig. 9 shows a graphical representation (volcano plot) of the differential analysis suggesting that our cutoff is somewhat conservative and indicates several significantly differential genes with low fold-change that are not discussed further in this report. It is clear that there are substantial differences in gene expression between infected and control samples. Our chosen fold-change boundaries indicated that 17 transcripts changed significantly (P < 0.05 with FDR filtering) between the uninfected and N. caninum-infected locusts. Of these transcripts, 10 increased >1.5-fold and 6 decreased >−1.5-fold (Table 1). Of the up-regulated transcripts to which a function could be assigned, one transcript was associated with signal transduction and autophagy. Other functional categories that included down-regulated transcripts were developmental processes and metabolism. The identification of these DE transcripts could provide a pipeline for promising targets to test subsequently in mice.

Table 1 List of the top 10 differentially expressed genes with the highest fold differences between Neospora caninum-infected locusts and uninfected controls.

Probeset ID	Gene symbol	Gene function	p-value	Fold differencea	
Upregulated gene expression in association with infection.
Genes with increased expression in brain of infected locusts vs. non-infected locusts	
1628992_at	CG34253	—	0.0184845	1.53941	
1623064_at	—	—	0.0115445	1.55291	
1626577_at	CG15556	—	0.0142076	1.56118	
1626033_at	Atg9	Autophagy-specific gene 9	0.00543545	1.55952	
1626376_at	ptip	CG32133	0.0333639	1.60544	
1635717_x_at	—	—	0.0410724	1.60703	
1629711_at	CG10597	—	0.00069658	1.61358	
1640991_at	CG32091	—	0.0186437	1.62449	
1632924_at	—	—	0.00142772	2.43486	
1635192_at	svp	seven-up	0.0317402	3.22575	
Downregulated gene expression in association with infection. Genes with decreased expression in brain of infected locusts vs. non-infected locusts	
1625538_at	Jon66Ci	Serine-type endopeptidase
activity	0.0163373	−1.5279	
1640434_at	mos	—	0.0376856	−1.55319	
1636202_s_at	ImpL2	Ecdysone-inducible gene L2
InR signalling pathway	0.0121551	−1.58749	
1636412_at	Mical	—	0.00149258	−1.60216	
1630146_at	—	—	0.0101685	−1.65487	
1634370_a_at	Panner (Pnr)	Zinc-finger transcription
factor of the GATA family
(involved in several
embryogenesis processes and
imaginal development)	0.0171417	−1.72354	
Notes.

a mRNA expression relative to uninfected controls, as determined by microarray analysis, with 3 locusts per group. No difference between groups, 1.00. Genes with unknown function are indicated by (—).

Hierarchical clustering and PCA identifies distinctive molecular phenotypes in infected vs. control locusts

DNA microarray expression profiling demonstrated significant differences in gene expression between infected and control locusts. From the 17 statistically-significant probes that represented multiple genes, a hierarchical clustering heat map revealed two distinctive patterns that closely correlated with the two study groups (Fig. 10).

Discussion

Previously, N. caninum was known to circulate between mammalian and bird species (Mineo et al., 2011) only and considered limited by a “species barrier”, which might depend on differences in the parasite virulence traits with regard to host animal physiology. Herein, we demonstrate for the first time that a strain of N. caninum (Nc-Liverpool) was able to infect the invertebrate desert locust with induction of signs of illness and death. The progression to a lethal N. caninum infection in this non-natural host may result from the cumulative deleterious effects of both direct and indirect consequences of parasite infection that lead to accelerated destruction and compromised host immune regenerative capacity. N. caninum did not elicit a typical mortality dose response curve after intra-hemocoel infection and the reason for this is not known. However, the lack of dose-dependent mortality can be explained by the existence of a threshold effect that occurs with relatively moderate number of parasites (103 to 104), after which large differences in parasite number make little difference. It is also possible that immune response might have been elicited against this virulent strain to control parasite dissemination and interfered with the fatal outcome following increasing infectious challenge doses of the parasite. Doses from 105 to 106 parasites might seem high for the establishment of an infection for downstream analysis. On the other hand, infection with less number of parasites (<1,000 tachyzoites) didn’t seem to cause any effect on the locust’s sickness, mortality, or body weight. For these reasons, a dose of 104 was chosen for subsequent experiments, due to the reasonably long mean time of death after infection, which allows more time for data collection but is pathogenic enough to determine the effect of infection.

Also, there was a reduction of body weight and fecal output in response to infection, but not in a dose-dependent manner. This is most likely due to the use of pooled rather than individual fecal samples. The possibility that this was caused by changes in the diet of locusts was ruled out by dietary standardization. Parasites disappeared from the hemolymph of the locusts after 2 day PI. Infected locusts showed reduced mobility and sickness. The possible risk due to the manipulation of the locusts during weighing or collection of hemolymph was ruled out because these handling issues didn’t seem to affect the control groups. Mock-infected group and environmental control locusts group, apart from the death of two and one locusts, respectively, appeared normal. The study of locusts’ behaviour highlighted the difficulties associated with the behavioural assay. Many of the tested variables are not independent of one another (e.g., walking vs. leg movement vs. inactivity). It could have been more useful to analyze the amount of time locusts spent doing each activity, rather than the number of bouts. Also, the assay has not been standardized to control for the time of day. Locusts show a certain amount of diurnal variation in activity levels that may have contributed to the variability of the assay results. Following the intra-hemocoel inoculation of N. caninum into locusts, N. caninum migrated to the brain within 24 h accompanied by induced histological alterations and infection-specific inflammatory responses, suggesting that the insect immune response is recognising and responding to the parasites. However, because S. gregaria is not a natural host of N. caninum in the field, it unlikely that locust is able to mount a N. caninum-specific response. It remains to be investigated if histopathological changes are N. caninum-specific or inflammatory response to the parasite invasion.

Previous studies in mice (Collantes-Fernández et al., 2004), wild rodents (Ferroglio et al., 2007), dogs (Peters, Wagner & Schares, 2000), or calves (Kritzner et al., 2002) showed that the parasite DNA and/or lesions could be detected in the brain as well as other tissues, such as muscle, heart, liver, spleen, lung, and pancreas. In the present study, although tissue tropism was evident by the presence of parasite DNA in the brain (Fig. 5), it was interesting to note that all examined non-brain tissues did not show evidence for the presence of the parasite, suggesting the brain as the preferred target organ in experimental N. caninum infection in locusts. An alternative explanation is that the parasite is strictly neurotropic and its absence in other tissues is due to the lack of significant neuronal tissues in other organs in the locust, unlike in vertebrates. These findings are different to an earlier report where the protist Acanthamoeba organisms were able to invade different organs of the infected locusts and did not exhibit any tissue-specific preference (Mortazavi et al., 2010).

Two developmental stages, a tachyzoite stage and a bradyzoite (cystic) stage, are known to occur during N. caninum infection in the vertebrate animals (Peters, Wagner & Schares, 2000; Kritzner et al., 2002; Collantes-Fernández et al., 2004; Ferroglio et al., 2007). In the present study, only tissue invasion by tachyzoites, commonly associated with acute phase of infection with N. caninum, was able to be established in the locusts. One might argue that the demonstration of the occurrence of the two life cycle forms of N. caninum within the locust might have made it more representative to the natural (or vertebrate) host. However, this is a new in vivo approach to culturing N. caninum and the first report of an establishment of N. caninum infection in an invertebrate host. The availability of this experimental invertebrate host opens an avenue for N. caninum research in defining determinants of the parasite virulence, and host roles in disease pathogenesis along with a reduction of the number of vertebrate animals used in research. The use of the invertebrate locust has proven valuable to discriminate molecules participating from both sides of the host-parasite interaction (Siddiqui et al., 2011). For example, neuropathogenic E. coli K1 pathogenesis within both locust and mammalian systems exhibited remarkable similarities in producing bacteremia leading to bacterial invasion of the central nervous system and has been shown to be dependent upon several common established virulence factors, such as LPS, OmpA, FimH, and CNF1 (Khan & Goldsworthy, 2007; Mokri-Moayyed, Goldsworthy & Khan, 2008). Assuming a correlation between the virulence of N. caninum in locusts and in vertebrates N. caninum-induced killing of the locusts can be exploited as an assay system to screen for neuropathogenesis virulence-attenuated mutants of N. caninum or in preclinical testing of interventional strategies to interfere with the growth of N. caninum tachyzoites.

An important result was the recovery of viable N. caninum tachyzoites from the brain of experimentally infected S. gregaria for up to 17 days PI. The isolation of a viable N. caninum isolates from the brain of infected S. gregaria provided an opportunity to examine whether passage of the parasites through a totally non-natural host induces phenotypic and/or genotypic changes in these parasites. Thus, the phenotype, genetic and chemical profile of N. caninum isolate derived from S. gregaria and the original isolate were compared. S. gregaria-derived isolate exhibited similar growth pattern to the original isolate (unpublished data). The two isolates were subjected to nucleotide sequence analysis of the Nc5 gene and compared to each other and to previously described N. caninum sequences available in Genbank database. PCR-sequencing analysis revealed that both parasite isolates are identical at least with respect to this gene. Due to the multi-copy nature of Nc5 gene direct sequencing of PCR products of this gene may not be that informative. Alternatively, cloning prior to sequencing followed by careful alignments of cloned sequences is a more sensitive approach to distinguish between all variants of the gene, and can enable the recovery of information useful for estimation of genetic diversity between N. caninum isolates. Minor, non-significant differences were found between the two isolates by using Raman spectroscopy. A large amount of spectral data obtained from the two isolates analysed by multivariate analysis did not reveal two distinct clusters, identifying only a few chemical metabolites that were different between the two parasite isolates (Fig. 7).

Figure 7 Comparative profiling of chemical structure of tachyzoites of Neospora caninum from locust-derived and original isolates using Raman spectroscopy.

(A) Comparative Raman spectra of tachyzoites of N. caninum culture-derived (blue) and locust brain-derived isolates (green) in the region from 700 to 1,700 cm-1. (B) Principal component analysis score plots in the plane of principal components 2 vs. principal component 1 for samples tested. Each dot represents a chemical molecule and the dots are colored according to the biological group the sample belongs to. Blue dots indicate N. caninum culture-derived isolate and green dots indicate locust brain-derived isolate. Minor (green dots at the top), but non-significant differences were present between the chemical profile of each isolate. Raman spectra of locust-derived isolate are less scatter (i.e., less variation in their structures) compared to spectra of culture-derived isolate.

One of the objectives of this study was to test whether passage of the parasite in a non-natural host can affect the parasite’s virulence and neurotropism. Such evidence was acquired by injecting parasites derived from brain of S. gregaria, via a cell culture bioassay, into new S. gregaria hosts. The locust derived isolate resulted in successful infection of 100% of the newly infected hosts, indicating that this parasite has the ability to adapt to changing physiological environment and retained its capacity to invade the brain in newly infected S. gregaria. Studying the implications of serial passages of N. caninum in S. gregaria on the parasite’s virulent activity and the evolution of the infection in the locusts are among the future directions of the work. Host adaptation is a well-known strategy to improve the infectivity of a pathogen in a non-natural host. However, it risks the concomitant disadvantage of biasing the natural tropism of the pathogen. Interestingly, the tissue tropism of N. caninum was not alerted after infection of locust. These results indicate that the current definition of the species barrier, which has been based on host specificity, needs to be reassessed for N. caninum. The possibility of N. caninum crossing the species barrier and infecting hosts of diverse phylogenetic backgrounds will have important evolutionary implications.

Fundamental to the understanding of the complex and multifaceted interactions between the parasite and locusts’ brain was the study of global locust’s response to infection by using high-throughput approaches, such as lipidomic and transcriptomic, validated for use in other species. Measuring the types and the abundance of lipids in brain from healthy and infected locusts using GC has provided a baseline lipid profile in S. gregaria brain and the subsequent response to N. caninum infection. The multi-variate prediction models generated using BioHEL were able to discriminate with good accuracy (98%) between the FA profiles of the infected and control samples (Fig. 8). Moreover, by restricting the models to use only small subsets of two, three and four fatty acids we identified that the minimal subset of FAs with the highest prediction capacity was that of size three (C16:1, C18:2n6c and C18:3n3). It is worth noting that C16:1 and C18:2n6c was always present in the best panels of FAs of sizes two, three and four. The idea that fatty acid metabolism is influenced by Neospora represents just one example where lipidomic analysis has provided new hypotheses to explore. This N. caninum-specific alteration in fatty acids is a characteristic also observed in a previous study based on transcriptome analysis in mice infected with N. caninum (Ellis et al., 2012).

Figure 8 Heat map of differentially expressed lipids in locust brains.

Unsupervised two-dimensional hierarchical clustering of the 7 fatty acids that showed fold change differences between infected locust groups (n = 5 locusts) and their adjacent control daily for 5 days post infection with Neospora caninum. The heat map of differentially expressed lipids based on clustering is shown in the figure. Each column represents a lipid species and each row represents a locust group. Red colour indicates lipids that were upregulated and yellow color indicates lipids that were downregulated. Orange indicates lipids whose level is unchanged in infected locust’s brain as compared to normal. A significant discriminative power between the infected and control samples of the locust’s brain was evident. Samples are identified by a three-part code: “F/C (infected/controls)”. “Time point”. “Replicate number”. Fatty acids are reordered after applying a hierarchical clustering to their profiles. Hierarchical clustering of the rows and columns highlights groups of significantly correlated infection and lipids.

Figure 9 Volcano plot representation of the microarray data showing both significantly expressed transcripts and magnitude of change.

Negative log10 p-value on y axis indicates the significance of each gene, and the fold change (log base 2) mean expression difference on the x axis. Each gene is represented by a dot. Data are representative of three hybridizations per group.

Figure 10 Hierarchical clustering of significantly expressed genes of three infected vs. three control locusts.

The heat map shows two relatively distinct clusters of highly differentially expressed transcripts obtained from pairwise comparison between infected vs. control locust groups. Each row represents each sample tested and each column represents a single probeset (gene). On the hierarchical tree at the left side of the diagram, the upper half (red) indicates the control samples and the lower half (orange) indicates the infected samples. Relative gene expression is color represented: red is higher-level expression relative to the sample mean, blue is relatively lower-level expression, grey is no-change. The 11 probesets/genes in the upper right quadrant of the cluster map are genes that decreased upon infection relative to the control samples (shown in the lower right quadrant). The 6 probes/genes in the left upper quadrant were genes that were increased in control samples relative to infected samples (in the lower left quadrant).

We next sought to determine the genes specifically altered in the brain by the presence of the parasite. We investigated transcriptome variations in brains of both N. caninum-infected S. gregaria and non-infected control locusts at 24 h PI. This unbiased approach allowed us to categorize genes in both infected and non-infected brain tissues whose expression was altered by the presence of infection. This is the first study to show the benefit of CSH for an orthopteran (S. gregaria) species using a whole genome oligonucleotide microarray of the fruit fly Drosophila melanogaster. In line with previous studies in mammalian (Nieto-Díaz, Pita-Thomas & Nieto-Sampedro, 2007) and avian hosts (Crowley et al., 2009), our CSH analysis using non-vertebrate insect species did not affect the reproducibility of the hybridization data from Affymetrix GeneChips. This analysis provided insights into the first global transcriptional response of locusts to N. caninum infection. Ten locust genes associated with the immune response and with a variety of cellular pathways were identified as up-regulated >1.5-fold and 6 as down-regulated >−1.5-fold 24 h after infection (Table 1). Most of the differentially expressed genes were not annotated, so their function is unknown. However, one of the upregulated genes is Atg9, which encodes for a transmembrane autophagy-related protein and has been shown to induce c-Jun N-terminal kinase (JNK) signaling and autophagy in response to oxidative stress in Drosophila (Tang et al., 2013). In mammalian cells, mAtg9 plays an essential role in regulating oxidative stress-induced JNK activation. One of the down-regulated genes is the pannier gene, which encodes a zinc-finger transcription factor of the GATA family and is known to be involved in several developmental processes during embryonic and imaginal development in Drosophila. In agreement with this finding, gene expression analysis of N. caninum infection in mice revealed changes in the expression of genes associated with mammalian development, embryogenesis and fatty acid metabolism (Ellis et al., 2012). Even though a few differentially expressed genes were found to be statistically significant our CSH experiment provides proof-of-principle of a transcriptomics workflow for investigating how locusts’ brain gene expression is modulated due to infection. Transcriptomic data obtained from locusts in the present study in addition to the S. gregaria expressed sequence tags (EST) data from the locust CNS (Badisco et al., 2011) represent an important source of information that will be instrumental in further unraveling the underlying mechanisms of brain dysfunction in locusts in response to infection with N. caninum or other neuropathogens.

Transcriptome profiling analyses during N. caninum infection in mice have been published (Ellis et al., 2010; Ellis et al., 2012). These studies revealed major changes of gene expression patterns depending on factors, such as N. caninum strain, the mouse type and time post infection. Given these facts, it is not surprising to observe differences in gene expression between the invertebrate locusts and the mammalian animals. Interestingly, cataloguing changes in S. gregaria host gene expression in response to N. caninum infection identified potential molecular processes associated with parasite colonization of locust brains. By looking more closely at the biological significance of some of the changes in gene expression using RT-qPCR and extending over a time course of the infection it can be possible to establish further the significance of the differential expression results obtained in our transcriptomic analysis. The alteration of gene expression may be a response to N. caninum infection per se or may be a component of disease pathogenesis. Subsequent studies should investigate the influence of N. caninum infection vs. virulence on gene expression. To determine which of the identified genes correlated with virulence rather than just N. caninum infection; various N. caninum isolates, such as the naturally attenuated Nc-Nowra, and NCts-8 (relatively avirulent) and its wild type (NC-1) isolate with different virulence and cystogenic capability should be utilized.

Invertebrates confer additional advantages compared to vertebrate models, since they are less expensive, easy to obtain, maintain and handle experimentally and can facilitate the development and testing of new treatment/preventive strategies. The importance of invertebrate animals for the study of fungal pathogenesis has been reviewed recently (Desalermos, Fuchs & Mylonakis, 2012). Studies over the past few years have also demonstrated that S. gregaria can be used as an alternative to mammalian models of infection; particularly to investigate host-pathogen relationship (Khan & Goldsworthy, 2007; Mokri-Moayyed, Goldsworthy & Khan, 2008; Mortazavi et al., 2009; Mortazavi et al., 2010). Our study provides a new opportunity for testing the feasibility of locusts as an alternative model of protozoal infections. However, there are still many questions to answer, including how N. caninum is able to reach and persist in the locust brain. Whether locusts provided a permissive environment to N. caninum or the parasite possesses a high affinity to brain tissue of S. gregaria, is still unknown. There are many possibilities for future studies utilizing this invertebrate host. It could be interesting to assess if locusts could act as paratenic host/reservoir of the infection. Also, it is important to determine, using a quantitative PCR, the amount of parasites able to cross the “brain blood barrier”. Sequencing a 337-bp fragment provided a preliminary evidence of the genetic similarity between locust-derived and original (culture-maintained) isolates; this short segment of the DNA is not enough to determine genotypic changes, more sensitive methods, such as SNP analysis should be pursued in subsequent studies. Finally, study of the possible change in the protein expression pattern could be more informative than the Raman spectrometry analysis.

Supplemental Information

Figure S1 Injection of Neospora caninum tachyzoites into the hemocoel of Schistocerca gregaria

Click here for additional data file.

Figure S2 Collection of hemolymph from locusts

Hemolymph was collected by insertion of a pipette tip through the locust arthrodial membrane at the base of the walking appendages.

Click here for additional data file.

Figure S3 Immunofluorescent detection of Neospora caninum tachyzoites

Immunofluorescence micrographs of human brain microvascular endothelial cells (HBMECs) 4 hrs after infection with N. caninum strain isolated from locust brain (A) or original strain (B). Infected HBMECs are immune-labelled with primary monoclonal mouse anti-NcSAG1 antibody that recognizes surface antigen of the parasite and secondary goat-anti-mouse IgG FITC conjugate (green) and nuclear DNA stained with propidium iodide (red). Scale bars, 10 µm.

Click here for additional data file.

Figure S4 Transmission electron micrographs (TEM) showing the ultrastructure of Neospora caninum

TEM of human brain microvascular endothelial cells (HBMECs) 24 h after infection with N. caninum strain isolated from locust brain (A) or original strain (B). Abbreviations: host cell nucleus (n) and rhoptries (rop). Arrows points at parasitophorous vacuole, which encloses a number of tachyzoites (Ta). Bars in A = 3 µm and b = 2 µm.

Click here for additional data file.

Figure S5 Principal component analysis (PCA) of infected vs. control locust brain samples

PCA resulted in two relatively distinct components of three infected (blue) and three uninfected samples (red).

Click here for additional data file.

Table S1 Prediction capacity of the multi-variate prediction models trained from the lipid profiles of the infected and control locust brains. Accuracy of each model was estimated using leave-one-out cross-validation

Click here for additional data file.

Supplemental Information 1 Locust model of infection

Click here for additional data file.

The authors wish to thank Professor Lord Sandy Trees for providing N. caninum (Nc-Liverpool) strain, Naveed Khan for providing comments on the experimental infection of the locusts, Ekramy Elmorsy for his help with some of the statistical analysis, and Pablo Fainberg for his help with lipid extraction. The support of the University of Nottingham’s High Performance Computing cluster is greatly appreciated.

Additional Information and Declarations

Competing Interests

Author Contributions

Ethics

Microarray Data Deposition

The authors declare there are no competing interests.

Mamdowh M. Alkurashi performed the experiments, analyzed the data, prepared figures and/or tables, reviewed drafts of the paper.

Sean T. May, Kenny Kong and Jaume Bacardit analyzed the data, contributed reagents/materials/analysis tools, prepared figures and/or tables, reviewed drafts of the paper.

David Haig contributed reagents/materials/analysis tools, reviewed drafts of the paper.

Hany M. Elsheikha conceived and designed the experiments, performed the experiments, analyzed the data, contributed reagents/materials/analysis tools, wrote the paper, prepared figures and/or tables, reviewed drafts of the paper.

The following information was supplied relating to ethical approvals (i.e., approving body and any reference numbers):

This study was reviewed by the University of Nottingham (UK) School of Veterinary Medicine and Science (SVMS) Ethical Review Committee. The Committee reviews all research studies involving School personnel and is chaired by Professor David Haig. The committee doesn’t provide an approval number, but the committee passed this study as good to proceed, not requiring any further ethical review as it involved invertebrates. FELASA guidelines as outlined in ‘principles and practice in ethical review of animal experiments across Europe (2005)’ and UK guidelines on the use of invertebrates in research were followed.

The following information was supplied regarding the deposition of microarray data:

European Bioinformatics Institute (EMBL-EBI): ArrayExpress accession E-MTAB-3132.

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
