# Peer review of "Susceptibility to experimental infection of the invertebrate locusts (Schistocerca gregaria) with the apicomplexan parasite Neospora caninum"

_PeerJ, doi:10.7717/peerj.674_

## Round 0.1 · original submission · Major Revisions

This manuscript describes the experimental infection of desert locusts, Schistocerca gregaria with the vertebrate pathogen Neospora caninum. I thank the Authors for submitting this manuscript to PeerJ. I recommend the Authors to address the changes suggested by two reviewers. It will certainly improve the quality of the manuscript. I recommend major revisions for this submission.

Reviewer 1 ·

Basic reporting

The article is well written with a couple of minor points described below:

1. L82 Ref needed for control of N. caninium by innate immune system.
2. L85 What are these phylogenetically different hosts? To my knowledge all hosts of N. caninium, definitive or otherwise, are birds or mammals, between which there is very high molecular and functional conservation of the immune system. Please clarify this point.
3. L172 change 'pi' to 'post infection (PI)'
4. L230 Should be 'dead' not 'died'.
5. L250, 281 etc. No need for italics.
6. L511, 529 etc. No need to refer back to figures.
7. L856 Better description of histology required for reader to interpret. Perhaps arrows could be added to point out relevant features.
8. L507 And bird species (e.g. Mineo et al., 2011)

Experimental design

The experimental design is generally good with following specific problems/comments:

1. Could the authors comment on whether the locusts were the solitary or gregarious morphotypes (from the images provided they look gregarious) and that they were uniformly one or the other. Although outside of the purview of this manuscript it would interesting to know the differences in the response to N. caninium infection between morphotypes.
2. L155, 280 Can locusts be sacrificed by cervial dislocation when they don't have a neck?
3. L422 Why is this data not shown?

Validity of the findings

1. L355 Lower doses (10^0, 10^1, 10^2) would be very useful and it sound as if the authors may these data (L523) and if so there is no reason not to present them. As it stands there is no evidence that locusts mount any sort of immune response; survival is not changed over three orders of magnitude of inoculum and it appears that N. caninium can proliferate freely.

2. In relation to point 1 above: Given the significant mortality with 10^3 it is interesting that surviving animals have normal body weight. Is this evidence of these animals were controlling the infection?

3. L370 This very interesting; Is it possible they are switching morphotype back to solitary form!? Do you have an example image?

4. L548 An alternative explanation is that the pathogen is always neurotropic and its absence in other tissues is due to the lack of significant neuronal tissues in other organs in the locust unlike in vertebrates.

·

Basic reporting

Please see general comments below

Experimental design

Please see general comments below

Validity of the findings

Please see general comments below

Additional comments

In this paper, the authors describe the experimental infection of desert locusts (Schistocerca gregaria) with the vertebrate pathogen Neospora caninum, a causative agent of spontaneous abortions in livestock. Overall, I felt that the manuscript was well written. The Introduction was concise and contained the appropriate background information. The Materials and Methods were detailed and described the use of an impressive range of assays to determine the effect of N. caninum infection upon S. gregaria. However, I did feel that the Discussion could be improved by making it more concise. I have the following specific comments/suggestions:
Line 123) Were the locusts used in experiments male, female or a mixture of both? This information should be specified. If both males and females were used, were there any sex specific differences observed in the results of the different assay? Were the number of males and females used balanced?
Line 212-213) Please report the final concentration of the reagents, rather than the stock concentration and the volume.
Line 226) Consider changing “measured” to “detected”.
Line 339-343) I was a bit unsure of what was actually done here. It would be good to add a sentence or two that explains more clearly what these steps actually do – i.e. how did these methods enable you use the Drosophila microarray chip to assay locust gene expression? What is the theory behind this approach?
Line 363-370) I think that there are some limitations associated with the behavioural assay. Many of the variables that were measured are not independent of one another – e.g. walking vs leg movements vs inactivity. This is not necessarily a problem, but I do think it means that the data needs to be analysed in a particular way. I would suggest that the authors try analysing the data using logistic regression (e.g. DOI: 10.1111/j.1469-185X.1999.tb00038.x), provided that it meets the assumptions of this kind of statistical test. It might be more useful to analyse the amount of time spent doing each activity, rather than the number of bouts (if this information was recorded). Another thought that I had was whether the assays were standardised in any way to control for the time of day? It is my understanding that locusts show a certain amount of diurnal variation in activity levels that may have contributed to the variability of the assay results.
Line 376 & 401) Are there any statistics available to support these claims?
Line 407) “parasite-specific product” – was the PCR product sequenced to confirm that corresponded to the targeted N. caninum sequence?
Line 410) Caution should be exercised when discussing differences in the amount of PCR product produced, as end-point PCR is only semi-quantitative.
Line 427-428) I think that the discussion of the genotyping results should be made more precise (here and elsewhere). As is, it sounds like the two N. caninum isolates are (completely) genetically identical, but there is insufficient evidence to support this claim, as only one region of the genome was sequenced. It would be more reasonable to state that Nc5 has been shown by previous studies to be a variable region and as such, might be useful as a marker for detecting genetic differentiation between different isolates. On this note – were the PCR products cloned before sequencing? As single isolates can contain multiple, Nc5 variants (DOI: 10.1007/s004360100463), directly sequencing PCR products may not be all that informative.
Line 498 – 504) This information should be placed in the figure legend, not in the main text.
Line 523 – 528) Is there any information available in the literature about the dose required to induce illness in vertebrate hosts? That is, how biologically relevant is the dose used to infect the locusts?
Line 541) If S. gregria is not a host of N. caninum in the field, then I would think it unlikely that S. gregria is able to mount a N. caninum–specific response?
Line 557 – 559) Do locusts live long enough for a cystic phase to occur?
Line 564 – 567) I think that it should be noted that this assumes that there is a good correlation between the virulence of N. caninum in locusts and in vertebrates.
Line 624 – 626) Please state the limitations of the data – i.e. most of the ten genes were not annotated, so their function is not known.
Line 660) Invertebrates are animals.
Line 673) Are there any examples from the literature that would support this claim?
Figure 2 & 3) I do not think a t-test is appropriate for this data, as t-tests do not include a correction for multiple corrections. It might be possible to use a repeated measures ANOVA.
Figure 4) Please include scale bars. I also suggest that the authors make clear which features of the images the reader should be focussing on.
Figure 8) Would it make more sense to cluster according to treatment/sample type, rather than fatty acid type?
Figure 10) Please make labels clearer.
Figure S4) It wasn’t clear to me where the parasite is.
Table 1) I think ‘increased’ is meant to be ‘decreased’ - half way down table, above Jon66Ci line.

---

## Round 0.2 · accepted · Accept

This MS is now accepted. Please take care of a few abbreviations, such as post infection (PI). Congratulations.

Reviewer 1 ·

Basic reporting

My previous comments have been addressed with the exception of the use of 'pi' for post infection. The first instance of this abbreviation (I think this is L176) needs to be written as post infection (PI) and all subsequent use need to be 'PI'. I still found a couple of 'pi's (e.g. line 176, 393, 405...).

Experimental design

My previous comments have been addressed.

Validity of the findings

My previous comments have been addressed.

Additional comments

My comments have been addressed with the minor exception mentioned above.